# Antibody–Drug Conjugates as an Emerging Therapy in Oncodermatology

**DOI:** 10.3390/cancers14030778

**Published:** 2022-02-02

**Authors:** Clara Esnault, David Schrama, Roland Houben, Serge Guyétant, Audrey Desgranges, Camille Martin, Patricia Berthon, Marie-Claude Viaud-Massuard, Antoine Touzé, Thibault Kervarrec, Mahtab Samimi

**Affiliations:** 1Team “Biologie des Infections à Polyomavirus”, UMR INRAE ISP 1282, Université de Tours, 31 Avenue Monge, 37200 Tours, France; clara.esnault31@gmail.com (C.E.); serge.guyetant@univ-tours.fr (S.G.); patricia.berthon@inrae.fr (P.B.); antoine.touze@univ-tours.fr (A.T.); thibaultkervarrec@yahoo.fr (T.K.); 2Department of Dermatology, Venereology and Allergology, University Hospital Würzburg, Josef-Schneider-Straße 2, 97080 Würzburg, Germany; Schrama_d@ukw.de (D.S.); houben_r@ukw.de (R.H.); 3Department of Pathology, Université de Tours, CHU de Tours, Avenue de la République, 37170 Chambray-les-Tours, France; 4McSAF, 1 rue Claude Thion, 37200 Tours, France; audrey.desgranges@mcsaf.fr (A.D.); camille.martin@mcsaf.fr (C.M.); marieclaude.viaudmassuard@mcsaf.fr (M.-C.V.-M.); 5Team Innovation Moléculaire et Thérapeutique (IMT), Groupe Innovation et Ciblage Cellulaire (GICC) EA7501, Université de Tours, 31 Avenue Monge, 37200 Tours, France; 6Department of Dermatology, Université de Tours, CHU de Tours, Avenue de la République, 37170 Chambray-les-Tours, France

**Keywords:** antibody–drug conjugates, oncodermatology, melanoma, skin squamous cell carcinoma, cutaneous T-cell lymphoma and Merkel cell carcinoma

## Abstract

**Simple Summary:**

Currently, the therapeutic arsenal to fight cancers is extensive. Among these, antibody–drug conjugates (ADCs) consist in an antibody linked to a cytotoxic agent, allowing a specific delivery to tumor cells. ADCs are an emerging class of therapeutics, with twelve FDA- and EMA-approved drugs for hematological and solid cancers. In recent years, tremendous progress has been observed in therapeutic approaches for advanced skin cancer patients. ADCs appear as an emerging therapeutic option in oncodermatology. After providing an overview of ADC design and development, the goal of this article is to review the potential ADC indications in the field of oncodermatology.

**Abstract:**

Antibody–drug conjugates (ADCs) are an emerging class of therapeutics, with twelve FDA- and EMA-approved drugs for hematological and solid cancers. Such drugs consist in a monoclonal antibody linked to a cytotoxic agent, allowing a specific cytotoxicity to tumor cells. In recent years, tremendous progress has been observed in therapeutic approaches for advanced skin cancer patients. In this regard, targeted therapies (e.g., kinase inhibitors) or immune checkpoint-blocking antibodies outperformed conventional chemotherapy, with proven benefit to survival. Nevertheless, primary and acquired resistances as well as adverse events remain limitations of these therapies. Therefore, ADCs appear as an emerging therapeutic option in oncodermatology. After providing an overview of ADC design and development, the goal of this article is to review the potential ADC indications in the field of oncodermatology.

## 1. Introduction

In 1909, the German chemist Paul Ehrlich [1] provided the first description of “targeted therapy”. He proposed the so called “magic bullet” concept, allowing the delivery of a therapeutic molecule to a specific target without affecting healthy tissues [2]. Currently, targeted therapies are used in daily oncology practice including oncodermatology. As such, tyrosine kinase inhibitors (TKIs) targeting BRAF and MEK proteins block the constitutive activation of the MAPK pathway in patients with BRAF-mutated melanoma [3], therefore reducing maintenance, development or dissemination of the cancer. Similarly, sonic hedgehog pathway inhibitors (sonidegib and vismodegib) act by inhibiting the smoothened protein involved in hedgehog signal transduction which plays a crucial role in basal cell carcinoma (BCC) development. 

Among targeted therapies, antibody–drug conjugates (ADCs) combine a monoclonal antibody (mAb) with a highly cytotoxic molecule, allowing its specific delivery to tumor cells [4]. In recent years, optimization of ADC technologies in order to increase their therapeutic performances and overcome their limitations as well as evaluation of their effects in combination with currently approved drugs have significantly expanded their use in oncology [5,6]. Indeed, twelve ADCs have been approved by the FDA for treatment of hematologic malignancies ((A) in Table 1) and solid tumors ((B) in Table 1). Moreover, the number of ADCs in development is steadily increasing, with 195 ongoing clinical trials (https://www.beacon-intelligence.com/, accessed on 8 December 2021). Overall, ADCs are currently being applied in all oncology areas including skin cancers.

The aim of the present review is to highlight current developments of ADCs in oncodermatology and identify potential opportunities in this field. 

## 2. ADCs in Oncology

### 2.1. Structure of ADCs

ADCs consist in a cytotoxic drug, i.e., a payload bioconjugated through a linker to mAb-targeting tumor cell antigens [7] (Figure 1 and Figure 2). The specific combination of these components determines both the therapeutic performance and safety of the ADC [4]. The following parts aim to provide a brief overview of the features of the different ADC components and how these can impact efficacy and safety. 

### 2.2. Targets

Target identification is a crucial point in ADC development [8,9]. Tumor-specific biomarkers previously used for diagnosis, as well as proteins overexpressed due to gene amplification or proteins involved in tumor aggressiveness [10] might be considered as potential ADC targets. Most ADCs bind to proteins expressed on the cell surface while targeting of intracellular tumor-associated proteins can also be achieved by using T-cell-receptor-like antibodies recognizing peptides in the context of presentation by MHC-I complexes [11]. In both cases, i.e., extra- and intracellular proteins, the specificity of the tumor antigen, antigen expression levels and antigen/ADC internalization [12] determine the performance of an ADC target [9].

To provide optimal payload delivery, high and homogenous expression of targeted antigens on the tumor cell surface is required [13]. By contrast, absent or low expression in healthy tissues [14] is expected in order to limit toxicities on physiologic cells (i.e., in order to avoid on-target off-tumor cytotoxicity). Nevertheless, several clinically approved ADCs such as brentuximab vedotin, targeting CD30 [15], polatuzumab vedotin, targeting CD79 [16], and inotuzumab ozogamicin, targeting CD22 [17], actually engage with ligands expressed by immune cells but still harbor therapeutic efficiency and acceptable tolerance. Moreover, most ADCs require internalization of the targeted surface antigen through receptor-mediated endocytosis and lysosomal degradation of the linker in order to deliver the payload [18]. One evasion mechanism that limits ADC performance is the downregulation of the targeted antigen expression, which obviously affects ADC binding to the target cells [9].

While antigens present on the cell surface are the most frequent candidates for ADC targets [19], secreted proteins/soluble antigens might additionally offer new opportunities by being more accessible. As a consequence, linker and drug have to be designed to be active in this context [14]. Some ADCs targeting non-internalizing soluble antigens, such as Tenascin-C splice variants or fibronectin, with maytansinoid and auristatins as payloads, have already demonstrated antitumor activity [20,21]. Importantly, targeting soluble antigens in the tumor microenvironment (TME) might impact not only tumor cells but also tumor stroma, extracellular matrix and blood vessels. Moreover, an association between bevacizumab, an antivascular endothelial growth factor (VEGF) monoclonal antibody, and an ADC has provided improved outcomes compared to a combination of bevacizumab with standard chemotherapy, in a phase Ib clinical study including patients with platinum-resistant ovarian cancer [22]. 

### 2.3. Antibodies

mAbs used in ADC structures are mostly of the IgG1 or IgG4 subtypes, and are chimeric or humanized in order to decrease their immunogenicity [4]. Both parts of the mAbs, i.e., the fragment antigen-binding (Fab) and fragment crystallizable regions (Fc region), contribute to the performance of an ADC. While affinity of a mAb for its epitope is determined by the Fab portion [10], the Fc part is determinant for mAb stability and interactions with immune cells such as antibody-dependent cell-mediated cytotoxicity (ADCC), complement-dependent cytotoxicity (CDC) and antibody-dependent cell-mediated phagocytosis (ADCP). To this end, the Fc part interacts with neonatal Fc receptor (FcRn) [23], a recycling receptor determining both IgG half-life and biodistribution [24], or complement (C1q) implied in CDC or Fc receptors on immune effector cells (FcγR), a family of receptors which may trigger an immune response upon Fc binding [25]. The ADC interactions with immune cells might either constitute an opportunity to induce an antitumor response, or represent a potential side effect, resulting in toxicity on immune cells [14]. Accordingly, to improve or avoid immune functions (ADCC, CDC, and ADCP), genetic engineering introducing mutations in the Fc domain of ADCs is used [14,26]. The Fc mutations aim to promote or impair interactions between ADCs and immune cells, Fc receptors or complement (C1q) in order to prevent or improve their activation, such as increased ADCC with ADC glyco-engineering (i.e., afucosylation) [27,28]. On the other hand, other Fc modifications have been proposed to increase ADC half-life via FcRn-mediated ADC recycling [9]. 

Additional factors related to the payload and linker affect ADC pharmacokinetics (PK) and pharmacodynamics (PD) [29]. Notably, ADC hydrophobicity is associated with shorter half-life in serum [30] and because most of the payloads are hydrophobic, attachment to a hydrophilic antibody remains a challenge [31]. To overcome hydrophobicity and ADC aggregation, the most common ADC modification consists in glycosylation, a naturally occurring post-translational modification influencing the solubility, antigenicity and stability of proteins [32]. PEGylation, the addition of polyethylene glycol (PEG) to the linker in order to mask payloads hydrophobicity and to improve the PK of ADCs (i.e., reduction in aggregation, improvement in solubility and half-life [31]), has also been proposed [32].

### 2.4. Payloads

While radionucleotides, toxins [33] and cytokines [34] have been suggested as ADC payloads [35,36,37], most preclinical and FDA/EMA-approved ADCs are bioconjugated with small cytotoxic molecules [38] with a broad range of structures and mechanisms of action [4]. Among these, duocarmycins and pyrrolobenzodiazepines (PBD) dimers induce DNA alkylation by linking alkyl residues to AT-rich regions or guanine, leading to cell apoptosis [38]. Cell death can also be induced by calicheamicins inducing DNA double-strand breaks [38]. Maytansine derivatives (including DM1 and DM4) [39] and auristatins (monomethyl auristatin E (MMAE), monomethyl auristatin F (MMAF)) [7] are poisons which inhibit microtubule polymerization by targeting tubulins, resulting in G2/M arrest and apoptosis. A third family of cytotoxic agents inhibit the topoisomerase enzymes, which control DNA structure changes. Topoisomerase inhibitors block the cell cycle ligation step, which generates DNA single- and double-strand breaks, leading to apoptotic cell death [40]. Trastuzumab deruxtecan and sacituzumab govitecan are both approved topoisomerase inhibitor ADCs [40]. Currently, auristatins, maytansine derivatives and PBD/PNU (anthracyclines family) represent 29%, 25% and 2% of the global active ADCs, respectively (https://www.beacon-intelligence.com/, accessed on 8 December 2021). Overall, MMAE remains the most frequent payload used for ADC development, involved in 19% of active ADCs for which the payload is disclosed (https://www.beacon-intelligence.com/, accessed on 8 December 2021). 

Beyond their direct cytotoxic tumor-targeted activity, highly membrane-permeable payloads, such as MMAE or calicheamicin, are able to induce a “bystander effect” [41] due to drug release from the primary target cells into the TME. The bystander effect enhances ADC efficacy by targeting not only stromal cells, which do not express the targeted antigen, but also antigen-negative tumor cells in cases of intratumoral heterogeneity [12] (Figure 2). By contrast, MMAF, another auristatin derivate, has low cell permeability due to its negative charge. While such properties limit the off-target toxicity of MMAF ADCs, their bystander activity remains low [12]. Regarding this limit, MMAF ADCs require high tumor expression of target antigen [42]. Nevertheless, a MMAF-conjugated ADC, belantamab mafodotin, was approved in 2020 by the FDA and the EMA for relapsed or refractory multiple myeloma treatment [43,44]. 

### 2.5. Linker

The linker is the ADC portion that connects antibody and drug. Depending on the payload and antibody properties, either cleavable or non-cleavable linkers are used. Cleavable linkers are used in combination with payloads being only fully active after lysosomal degradation (e.g., MMAE or calicheamicin) [38]. To prevent the cleavage of the linker in blood circulation resulting in unspecific cytotoxicity, the ADC linker has to be stable until the ADC reaches its target cell. To this end, cleavable linkers are usually sensitive to conditions specifically found in lysosomes (e.g., acidic condition or lysosomic proteases (cathepsin B, glycosidase, phosphatase)) or high intracellular glutathione concentrations [38] and thereby release the payload only after endocytosis is achieved, leading to death of the target cell [38] (Figure 2). 

Regarding non-cleavable linkers, the drug is active after internalization and enzymatic digestion of the antibody into the lysosome, allowing the release of an active metabolite, amino acid–linker–drug complex (e.g., trastuzumab emtansine) [7]. Non-cleavable linkers allow increased ADC stability, leading to reduced toxicity in non-target tissues [7]. Payloads such as MMAF or DM1 are classically used in this setting. 

### 2.6. Linker–Antibody Conjugation

The drug to antibody ratio (DAR), defined as the number of drug molecules attached to one mAb, is another important determinant for ADC performance [38]. The optimal DAR depends on the nature of the payload [45]. The paradigm that a DAR value of 4 is optimal for pharmacokinetics has been challenged by the recent approval of DAR 8 antibodies, in particular in light of improved hydrophobicity-masking technologies [46,47]. The technology connecting the mAb to the linker is crucial to obtain a homogeneous and controlled DAR (i.e., position of drugs and loading) [48]. Chemical linkers involving functional groups (i.e., bioconjugation head) are used to bind to mAb amino acids. First-generation bioconjugation technologies involved either N-hydroxysuccinimide (NHS) ester bioconjugation heads binding to lysine residues, or hydrazones, maleimides and thioethers binding to cysteines [49]. However, those bioconjugation heads led to the generation of heterogeneous ADCs, presenting a high variability of the DAR in the range of 0–8, and to multiple different linker positions [38]. 

More recently, site-specific conjugation, i.e., engineered cysteine residues, unnatural amino acids, or enzymatic conjugation through glycosyltransferases have been applied to obtain more homogeneous ADCs [7]. Indeed, site-specific conjugation of antibodies has been shown to improve the therapeutic index because it improves ADC pharmacokinetics by reducing the hydrophobicity of the linker–payload as well as prevent the release of payload in blood [48]. Moreover, site-specific conjugation with rebridging of antibody disulfide bonds results in stable and homogeneous ADCs with a controlled number and position of payload, and higher stability characteristics [50]. 

Interestingly, Van Geel et al. has shown that it is possible to link a payload to the native N-glycan residing at asparagine 297 and demonstrated in vitro and in vivo efficacy [51]. 

To conclude, ongoing optimization of ADC technology is likely to expand their application in oncology in the upcoming years, as illustrated below with ADCs currently approved or investigated in the field of oncodermatology.

## 3. ADCs in Cutaneous T-Cell Lymphoma

Cutaneous T-cell lymphoma (CTCL) was the first skin cancer for which a therapeutic ADC was approved. CTCLs are a group of extra-nodal non-Hodgkin’s lymphomas with primary cutaneous infiltration of malignant monoclonal T lymphocytes. Mycosis fungoides (MF) is the most common form of CTCLs, characterized by slow progression from patches to infiltrating plaques and eventually to cutaneous and extracutaneous tumors. Among CTCLs, Sézary syndrome (SS) is characterized by erythroderma, lymphadenopathy and systemic dissemination of malignant Sézary T cells [52]. Although rare, SS displays an aggressive course, with a median survival of 1 to 5 years [53]. In the case of advanced MF and SS, the only potentially curative treatment is allogenic hematopoietic stem cell transplantation [54]. CD30, a proliferation-promoting member of the TNF receptor family [54], is expressed in 12 to 23% of SS and MF cases [55]. Since CD30 expression is restricted in healthy human tissues to activated T, B and NK cells, it was chosen as a potential ADC target for treating advanced CTCLs [10] **(**Figure 3A). Accordingly, brentuximab vedotin (adcetris^®^), consisting of an anti-CD30 MMAE-conjugated ADC, with a cathepsin B cleavable linker, was developed (Table 2). Clinical benefit of brentuximab vedotin was demonstrated in CTCLs (stage IV SS and MF) with clinical objective response rates (ORR) of 70% [56,57]. Moreover, brentuximab vedotin achieved more durable responses (56% ORR lasting 4 months) than conventional treatments with methotrexate or bexarotene (12% ORR lasting 4 months) (Table 3). Consequently, brentuximab vedotin was approved by the FDA and the EMA in 2017 to treat CD30+ CTCLs after at least one line of systemic therapy. 

However, resistance to MMAE was observed upon brentuximab vedotin treatment [58]. To overcome this resistance mechanism, a DM1-conjugated anti-CD30 ADC is currently in development. The anti-CD30 antibody is conjugated to DM1, a tubulin inhibitor, through antibody lysines by NHS ester non-cleavable linker (Table 2). This ADC has shown similar antitumoral effects to brentuximab vedotin without acquisition of payload resistance mechanism; moreover, the payload-bioconjugation modifications may have safety advantages comparing to brentuximab vedotin, without retro-Michael instability of maleimide. This ADC is currently evaluated in an ongoing clinical trial of patients with recurrent or refractory CD30+ hematological malignancies (NCT03894150). 

Another approach includes the combination of brentuximab vedotin with other therapeutics in order to improve efficacy and/or circumvent resistance. For instance, combination with the PD-1 blocking antibody nivolumab is currently being investigated in two phase I/II trials [59] (NCT02581631 and NCT01703949). Similarly, a phase I study is currently investigating the association between brentuximab vedotin and romidepsin, a histone deacetylase inhibitor which is FDA approved as a monotherapy for CTCL patients (NCT02616965). 

Since patients treated with brentuximab vedotin do not achieve improvements in long-term outcomes, new CTCL targets were identified and new ADCs were developed. Inducible Co-Stimulator (ICOS) is a T-cell costimulatory receptor involved in the development of CTCLs. High ICOS expression in CTCLs suggested it as a suitable target to develop an anti-ICOS MMAE-conjugated ADC [60]. Interestingly, a comparison between brentuximab vedotin and an anti-ICOS MMAE-conjugated ADC in a preclinical model of a CTCL xenograft showed a longer overall survival of mice [60]. Recently, cell surface proteome analysis on CTCL cell lines identified cell surface heat shock protein 70 (scHSP70) as highly expressed compared with normal T cells [61]. Then, an anti-scHSP70 MMAE-conjugated ADC was developed, and in vitro comparison with brentuximab vedotin showed similar activity against MS and SS cell lines [61]. However, under hypoxic conditions, such as a tumor microenvironment, scHSP70 expression was higher and cells were obviously more sensitive to the ADC—this result suggests an advantage over brentuximab vedotin [61].

## 4. ADCs in Melanoma

While BRAF/MEK inhibitors or immune checkpoint inhibitors (ICI) have revolutionized the treatment of patients with metastatic melanoma, approximately 50% of patients still experience fatal outcomes. Although 70% of patients with metastatic melanoma respond upon treatment with a combination of BRAF and MEK inhibitors [62], acquired resistance is frequent, especially in those with high tumoral burden at baseline. Acquired resistance to MAPK pathway inhibition is frequently associated with reactivation of the MAPK pathway due to secondary genetic (NRAS mutation, BRAF amplification, and MEK mutations) or epigenetic changes (Akt amplification, loss of PTEN, amplification of HGF, RTK (receptor tyrosine kinase), PDGFRβ, and IGF 1R) [3]. Metastatic melanoma patients can be treated with ICI such as the anti-PD1 antibodies pembrolizumab or nivolumab as monotherapies [63] or nivolumab combined with ipilimumab, an antibody targeting CTLA-4 on T cells [63]. Overall, primary resistance occurs in 40 to 65% of metastatic melanoma cases either treated with targeted therapies, i.e., BRAF/MEK inhibitors or ICI [64], highlighting the need for new therapeutic options [65]. In this context, several ADC strategies targeting tumor cell membrane proteins, mostly tyrosine kinase receptors, or soluble proteins, have been developed in recent years (Table 4). ADCs developed against melanoma are presented in three parts depending on their target types (Figure 3B).

### 4.1. Membrane Protein as Targets

Glycoprotein-NMB (gpNMB) is a transmembrane glycoprotein involved in tumor growth, tumor invasion and metastasis. Moreover, gpNMB exerts a direct inhibitory effect on activated T cells, impairing antitumor immunity and allowing immune evasion [66]. Accordingly, high gpNMB tumor expression levels are associated with poor clinical outcome [66]. In melanoma, 87% of tumors demonstrate membrane expression of gpNMB [67,68]. By contrast, in normal cells, gpNMB is restricted to intracellular compartments, therefore suggesting gpNMB as a promising ADC target in melanoma [66]. Accordingly, glembatumumab vedotin, a gpNMB-targeted MMAE-conjugated ADC [69], resulted in a 39% ORR in a phase I/II study conducted among patients with advanced melanoma refractory to ICI and BRAF/MEK inhibition, with acceptable toxicity (NCT00412828) [70]. In line with the observation that gpNMB expression is induced by the inhibition of MAPK pathway [71], combination of glembatumumab vedotin with MAPK pathway inhibitors demonstrated a synergistic therapeutic effect in a melanoma mouse model [71]. 

High and homogeneous expression levels of HER3, a tyrosine kinase receptor, have been observed in 65% of cutaneous melanomas [72]. HER3 is a member of the HER/EGFR family, and heterodimerization of HER3 with other HER family members (e.g., EGFR or HER2) leads to the activation of the PI3K/AKT pathway [73]. Accordingly, HER3 expression promotes tumor growth, and its overexpression was associated with impaired survival in melanoma [8,73]. Moreover, BRAF/MEK inhibitors induce FOXD3 expression, a transcription factor triggering HER3 expression. Hence, upregulation of HER3 by FOXD3 can promote adaptive resistance to BRAF inhibitors [74]. As a consequence, co-targeting HER3 and BRAF/MEK appears as an interesting option to overcome therapeutic resistances in BRAF-mutated melanoma [74]. However, unconjugated anti-HER3 antibodies have only shown limited antitumor activity justifying the use of anti-HER3 ADCs [75]. Indeed, an anti-HER3 MMAF-conjugate exhibited complete and durable antitumor responses in xenograft mice models [76]. Another anti-HER3 topoisomerase I inhibitor-conjugated ADC [77] is evaluated as monotherapy in phase I/II clinical trials for melanoma (NCT02980341). 

Finally, Chen and collaborators identified the melanosomal protein, PMEL17, as a highly expressed marker restricted to melanoma cells. Accordingly, they generated an anti-PMEL17 MMAE-conjugated ADC which was cytotoxic in vitro and in vivo in a preclinical model [78].

### 4.2. Tyrosine Kinase Receptor

Endothelin B receptor (ET_B_R) is a member of the G protein-coupled receptor superfamily mediating tissue differentiation, growth, and repair, through the MAPK pathway [79]. ET_B_R has been reported to be overexpressed in metastatic melanoma compared to normal melanocytes [80]. ET_B_R signaling has been implicated in malignant transformation of melanocytes, suggesting it as an oncogenic driver for melanoma development [81]. Importantly, ET_B_R blockade results in inhibition of melanoma growth in vitro and in vivo [81], rendering ET_B_R as a possible target for an ADC [82]. Accordingly, an anti-ET_B_R MMAE-conjugated ADC is currently evaluated in a phase I clinical trial in patients with metastatic or unresectable melanoma (*n* = 53) (NCT01522664) [83]. Upon treatment, 32% of patients had stable disease beyond 6 months, irrespective of BRAF status [83]. Interestingly, in a preclinical model, combination of a single dose of ADC and daily intake of MAPK pathway inhibitors increased expression of ET_B_R and enhanced the therapeutic response [84]. 

Anexelekto (AXL) is a tyrosine kinase receptor frequently expressed in melanoma [85]. Its tumoral expression has been associated with a more invasive phenotype [85]. Antitumor activity of an anti-AXL MMAE-conjugated ADC was observed in a preclinical model, and again combination with BRAF/MEK inhibitors improved antitumoral cytotoxicity [86]. Importantly, a BRAF/MEK inhibitor-resistant but high-AXL-expressing tumor sub-population was killed by the anti-AXL ADC, while BRAF/MEK inhibition was specifically cytotoxic for the low-AXL-expressing tumor cells [86]. Following this promising preclinical study, a phase I/II clinical trial is ongoing to evaluate this ADC (called enapotamab vedotin) in solid tumors including melanoma (NCT02988817). Moreover, AXL could also be an interesting target for squamous cell carcinoma [87]. 

c-KIT is a tyrosine kinase receptor engaged by a ligand called stem cell factor. c-KIT is involved in regulation of apoptosis, differentiation and proliferation [88]. In cancer, c-KIT mutation or gene amplification results in uncontrolled proliferation and resistance to apoptosis [88]. High c-KIT expression is observed in 64 to 88% of melanomas [88]. So far, c-KIT-targeted tyrosine kinase inhibitors only showed limited clinical benefit in *Kit*-mutated metastatic melanoma, due to the emergence of secondary mutations [89]. Preclinical studies revealed a potent antitumor activity of an anti-c-KIT DM1-conjugated ADC on several c-KIT-positive solid cancers including melanoma and leukemia [90]. Nevertheless, in a phase I clinical trial (NCT02221505), acute hypersensitivity reactions triggered by degranulation of c-KIT-expressing mast cells led to the termination of the trial [90]. 

### 4.3. Soluble Target for ADCs

Non-internalizing ADCs are also investigated for melanoma treatment using an ADC targeting the galectin-3-binding protein (Gal-3BP), a metastasis-associated secreted protein [91] which is abundant in the TME [92]. Following treatment with anti-Gal-3BP ADC, complete tumor regression was achieved in a xenograft mouse melanoma model [91]. 

## 5. ADCs in Skin Carcinomas

ADC strategies have also been developed for treating skin carcinomas which are prone to display an aggressive course, mostly squamous cell carcinoma (SCC) and Merkel cell carcinoma (MCC). 

### 5.1. Squamous Cell Carcinoma

SCC is a common skin cancer frequently induced by cumulative exposure to ultraviolet radiation [93]. SCC prognosis is mostly favorable after complete tumor resection while poor outcome is observed at the metastatic stage, with no demonstrated benefit of conventional chemotherapies or EGFR inhibitors on patient survival [93]. Recently, one PD-1 inhibitor, cemiplimab, was shown to provide a 47% ORR in patients with advanced or metastatic cutaneous SCCs, with durable responses in 61% of them [94,95]. PD-1 inhibitors are currently recommended as first-line treatment of advanced SCCs which are not candidates for surgery or radiation therapy [96].

Several targets have been identified as candidates for ADC therapy in lung or cervical SCCs [97] (Figure 3C). Most of the studies evaluated ADCs in head and neck SCCs. Overexpression of tissue factor (TF)—a transmembrane protein activating pro-survival pathways [98]—has been associated with high metastatic potential and poor outcome in various solid tumors [99]. High TF expression levels are also associated with increased expression of VEGF, enhancing tumor angiogenesis [98]. Tisotumab vedotin, a MMAE-conjugated anti-TF ADC with a protease-cleavable linker ((A) in Table 5), demonstrated a 16% ORR (95% CI 10.2−22.5) in a phase I/II study with 147 patients with multiple solid tumors, including SCCs of the head and neck. The safety profile appeared manageable although 27% of patients had a treatment-emergent serious adverse event related to the drug, and one death from pneumonia possibly related to the treatment [100]. A phase II study is ongoing to determine the efficacy and safety of tisotumab vedotin after failure of first-line standard of care therapy, in solid tumors including head and neck SCCs (NCT03485209). 

Leucine-rich repeat containing 15 (LRRC15) is a marker of cancer-associated fibroblasts [101]. TGF-β, an immunosuppressive cytokine frequently expressed by tumor cells, induces LRRC15 expression in activated fibroblasts in the TME [102,103]. Targeting cancer-associated fibroblasts through LRRC15 in order to reduce immunosuppressive properties of the TME could overcome therapeutic resistances [104]. In line with the frequent LRRC15 expression in SCC cancer-associated fibroblasts (81% of cases) [101], an anti-LRRC15 MMAE-conjugated ADC, ABBV-085, triggered complete response in SCC xenograft (PDX) models [101]. Moreover, combination of this drug with an anti-PD1 antibody (mouse IgG2a, 17D2 clone) has shown potent activity in tumor models [101]. This ADC is currently being investigated in a phase I clinical trial including head and neck SCCs (NCT02001623) (Table 5A). Of note, LRRC15 is also expressed by melanoma tumor cells, which could constitute an additional application field for this ADC [101]. 

The major issue of ADCs targeting a molecule expressed by healthy tissues is illustrated by the outcome of the program development of bivatuzumab mertansine (Table 5A). This anti-CD44v6 DM1-conjugated ADC, CD44v6 being an aggressive variant of CD44, was assessed in four clinical trials including one enrolling head and neck SCC patients. After occurrence of a case of fatal toxic epidermal necrolysis together with other skin-related adverse events, clinical development of the ADC was discontinued [105]. This severe adverse event probably occurred due to CD44v6 expression by both SCC and normal squamous epithelium [105].

### 5.2. Merkel Cell Carcinoma

Merkel cell carcinoma (MCC) is an aggressive skin cancer induced either by UV light or the Merkel cell polyomavirus [106,107]. Until 2017, only cytotoxic chemotherapies, mostly platin salts and etoposide, were available for patients with advanced disease. In recent years, avelumab, an anti PD-L1 antibody, was shown to provide a 33% ORR in patients with metastatic disease, after failure of a first-line chemotherapy—most of these responses being durable. Avelumab was approved by the FDA and the EMA for treating patients with advanced metastatic MCCs [108]. Pembrolizumab was assessed in the first-line setting and was approved by the FDA in 2018 for treating advanced MCCs [109,110]. Although responses to ICI may be long-lasting, more than 50% of patients do not respond or develop secondary resistance [108].

CD56 has recently been suggested as a suitable target for antibody-dependent cellular cytotoxicity in MCCs, as it is expressed on the tumor cell surface by the majority of MCCs [111] (Figure 3C). Accordingly, an anti-CD56 DM1-conjugated ADC, lorvotuzumab mertansine (LM, also known as IMGN901), has been developed by covalently coupling the DM1 to the humanized anti-CD56 mAb, lorvotuzumab (huN901) (Table 5B). In a phase I clinical trial, this ADC demonstrated acceptable safety and tolerability in different CD56-expressing tumors [112,113]. Moreover, signs of clinical activity were especially detected in the four MCC-tested patients with two objective responses [112]. The following phase I/II clinical trial investigated the combination of carboplatine/etoposide/LM versus carboplatine/etoposide alone in patients with small-cell lung cancer [114]. This trial was first modified by a reduction in the dose of carboplatine/etoposide and finally discontinued because of safety concerns (NCT01237678) [114]. 

Recently, a new CD56-targeting MMAE-conjugated ADC, Adcitmer^®^, using a new bioconjugation approach has shown the control of MCC tumor growth in a mouse preclinical model [115].

Delta-like protein 3 (DLL3) is an inhibitory ligand of NOTCH receptors and is involved in neurogenesis during early embryonic development. DLL3 is upregulated in neuroendocrine tumor and has a minimal expression in normal tissues [116]. DLL3 expression was found in 91% of MCC patients [117] and high expression of DLL3 was associated with virus-positive MCCs [118]. Rovalpituzumab tesirine, an anti-DLL3 PBD-conjugated ADC [118], showed modest antitumor activity in patients with small-cell lung carcinoma treated in third lines [119] (NCT02674568), but such anti-DLL3 ADCs could represent a potential strategy for treating viro-positive MCCs. 

## 6. Strengths and Weaknesses of ADCs: Challenges and Perspectives

When compared to other cancer therapies, notably conventional cytotoxic treatment, the use of ADCs is a vectorized therapy with low off-target toxicity. Nevertheless, ADCs could also have some drawbacks, due to recognition of the target on non-tumor cells, leading to “on-target off-tumor cytotoxicity”, limiting the therapeutic window in which ADCs can be applied [14]. Moreover, unspecific linker cleavage can cause drug delivery without antibody recognition, resulting in systemic toxicity, which represents another important limitation of ADCs [120]. Improvements in the production process as well as modifications in ADC structures may counteract these constraints [14]. In order to obtain stable ADCs, engineering of the bioconjugation head might contribute to linker stability improvement [9]. As an example, maleimide residue either included in the linker or the bioconjugation head might react with free thiol of plasma, e.g., albumin, resulting in the so called “retro-Michael reaction”. Hence, this retro-Michael reaction induces an unspecific release of the linker and drug in the plasma, which not only impacts the efficacy of the ADCs but also leads to off-target toxicity [120].

Several resistance mechanisms to ADCs have been described such as target antigen downregulation or mutations, and impairment of lysosomal degradation pathways. Moreover, ADC payloads can be rejected from tumor cells though multidrug resistance transporter efflux [121]. For instance, loss of CD30 expression can be observed in cutaneous CD30-positive lymphoid neoplasm treated with brentuximab vedotin [58,122]. Trogocytosis, i.e., the extraction of the antibody–epitope complexes by monocytes may induce resistance to the ADC in CD30-positive lymphoma [122].

A wide number of ADC optimizations have been developed to improve ADC performance. A new class of therapy aiming to improve the tumor specificity of mAbs, named probodies, consists in the optimization of mAbs used for ADC technology [123]. These antibodies are modified in order to mask the Fab paratopes, thereby limiting their activity in healthy tissue. By contrast, in the TME, high protease activity removes the masking peptide, allowing paratope–epitope recognition. Enhancement of the therapeutic index by such probody-based therapy was demonstrated in an EGFR-overexpressing mouse model [123]. The use of probodies in ADCs to enhance tumor cell-specific delivery of drugs has recently been tested [124]. In this respect, an anti-CD166 probody–drug conjugate has shown on-target on-tumor specificity in a preclinical study on lung cancer [125]. 

ADC antitumoral effect could be optimized by the use of innovative payloads or a new bioconjugation strategy. As an example, trastuzumab emtansine is indicated as second-line treatment for HER2-positive metastatic breast cancer. Interestingly, a second anti-HER2 ADC, trastuzumab deruxtecan [126], with eight topoisomerase inhibitors conjugated to trastuzumab through a cleavable linker was recently approved by the FDA and the EMA—this ADC has shown higher efficacy than trastuzumab emtansine, demonstrating the involvement of a payload and bioconjugation strategy in ADC performance [126]. 

Among the challenges, ADC toxicity (i.e., thrombocytopenia, neutropenia or peripheral neuropathy) should be known and controlled, for example by dose modulation [127]. Moreover, unexpected combinatorial effects, in particular atypical ocular toxicities of bleomycin and brentuximab vedotin observed for Hodgkin’s lymphoma, should be considered for future combination strategies [127]. 

## 7. Conclusions

Several targets in skin cancers are currently being investigated as candidates for ADC therapy, which allows the targeted delivery of a drug. ADC is a growing class of therapeutics in oncology, including one ADC approved for cutaneous T-cell lymphoma. In preclinical and clinical trials, further ADCs are currently evaluated for the treatment of CTCLs, melanoma, SCCs or MCCs. Various combinations of ADCs with other therapeutics are investigated to overcome limitations of these drugs due to intrinsic or acquired resistance. Increased knowledge on critical ADC features, such as target, antibody, linker and drug choice, allows a steady improvement in ADC design and better management of side effects.

## Figures and Tables

**Figure 1 cancers-14-00778-f001:**
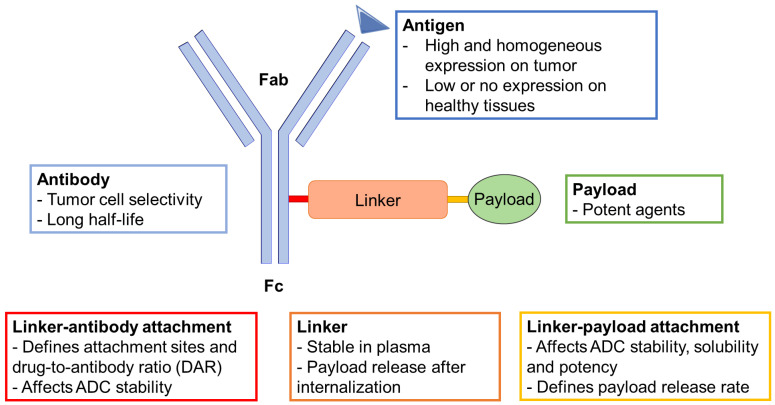
Design of an antibody–drug conjugate and recommended biological properties. Characteristic of antigen, antibody, linker–antibody attachment, linker, linker–payload attachment and payload are detailed. The antibody Fc part is implied in half-life and immunogenicity. The Fab part controls affinity and avidity to the targeted antigen.

**Figure 2 cancers-14-00778-f002:**
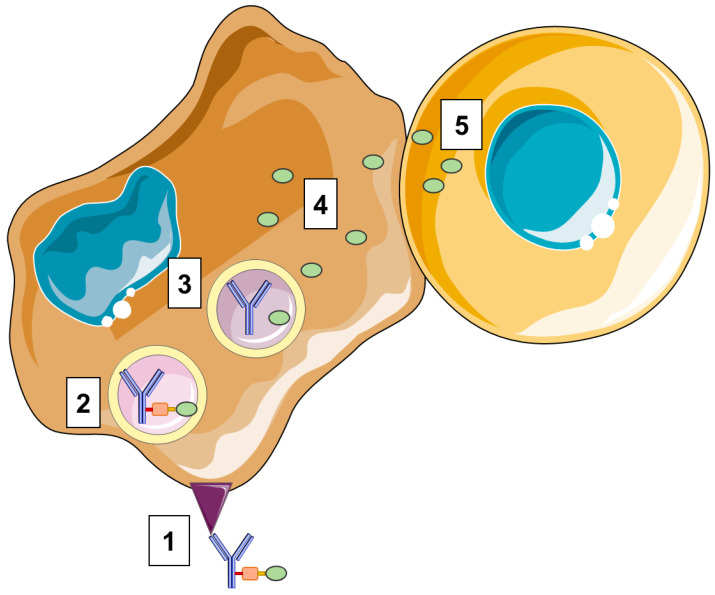
Overview of ADC mode of action. 1. Binding of the ADC to the antigen expressed by the cancer cell. 2. Internalization of the ADC. 3. Degradation of the linker or antibody inside the lysosome induces the release of an active form of the payload. 4. The payload exerts cellular toxicity depending on its mode of action. 5. A bystander effect can occur.

**Figure 3 cancers-14-00778-f003:**
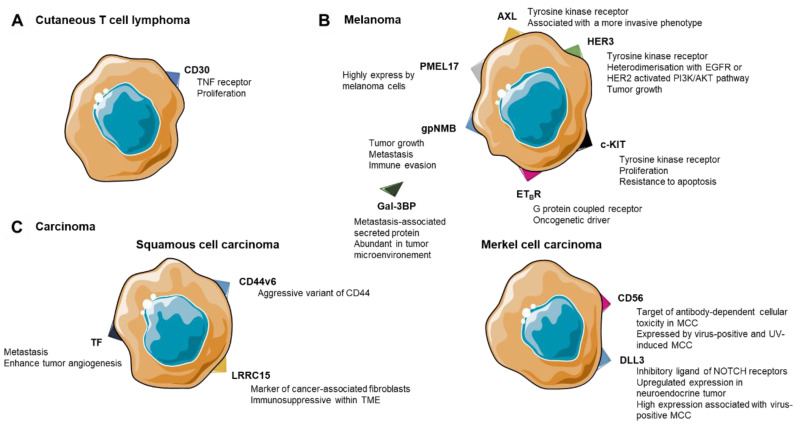
ADC targets in oncodermatologic indications and implication of antigen expression in cancer aggressiveness. (**A**) Cutaneous T-cell lymphoma. (**B**) Melanoma. (**C**) Carcinoma: squamous cell carcinoma and Merkel cell carcinoma.

**Table 1 cancers-14-00778-t001:** FDA-approved ADCs. A. Hematological malignancies. B. Solid tumors.

Hematological Malignancies							
**Commercial Name**	International Non-Proprietary Names (INN)	Target	Antibody Isotype	Bioconjugation Head (Antibody Amino Acid)	Linker	Drug (Therapeutic Class)	Indication
Adcetris^®^	brentuximab vedotin	CD30	Chimeric IgG1	Maleimidocaproyl (Cysteine)	Cleavable/proteolytic (cathepsin B)	auristatin	anaplastic large cell lymphoma + Hodgkin’s lymphoma
Polivy^®^	polatuzumab vedotin-piiq	CD79b	Humanized IgG1	Maleimidocaproyl (Cysteine)	Cleavable/proteolytic (cathepsin B)	auristatin	relapsed or refractory diffuse large B-cell lymphoma
Mylotarg^®^	gemtuzumab ozogamicin	CD33	Humanized IgG4	Acetyl butyrate (Lysine)	Cleavable/hydrazone	calicheamicin	CD33-positive acute myeloid leukemia
Beponsa^®^	inotuzumab ozogamicin	CD22	Humanized IgG4	Acetyl butyrate (Lysine)	Cleavable/hydrazone	calicheamicin	lymphoblastic leukemia
Lumoxiti^®^	moxetumomab pasudotox	CD22	N/A	N/A	Cleavable/proteolytic (furin)	*Pseudomonas* endotoxin A	hairy cell leukemia
Zynlonta^®^	loncastuximab tesirine-lpyl	CD19	Humanized IgG1	Maleimidocaproyl (Cysteine)	Cleavable/proteolytic (cathepsin B)	pyrrolobenzodiazepine dimer (PBD)	relapsed/refractory diffuse large B-cell lymphoma
Blenrep^®^	belantamab mafodotin	CD38	Humanized IgG1k	Maleimidocaproyl (Cysteine)	Uncleavable	auristatin	relapsed/refractory multiple myeloma
**Solid Tumors**							
**Commercial Name**	**International Non-Proprietary Names (INN)**	**Target**	**Antibody Isotype**	**Bioconjugation Head (Antibody Amino Acid)**	**Linker**	**Drug**	**Indication**
Kadcyla^®^	ado-trastuzumab emtansine	HER2	Humanized IgG1	Maleimidocaproyl (Lysine)	Uncleavable	maytansine	HER2/neu positive breast cancer
Padcev^®^	enfortumab vedotin	Nectin-4	Human IgG1	Maleimidocaproyl (Cysteine)	Cleavable/proteolytic (cathepsin B)	auristatin	locally advanced or metastatic urothelial cancer
Enhertu^®^	trastuzumab deruxtecan	HER2	Humanized IgG1	Maleimidocaproyl (Cysteine)	Cleavable/proteolytic (cathepsin B)	deruxtecan (topoisomerase inhibitor)	breast cancer HER2 positive after two or more lines of anti-HER2 therapy
Trodelvy^®^	sacituzumab govitecan	TROP-2	Humanized IgG1	Maleimidocaproyl (Cysteine)	Cleavable/hydrazone	topoisomerase inhibitor	metastatic triple-negative breast cancer
Tivdak^®^	tisotumab vedotin	Tissue factor	Human IgG1	Maleimidocaproyl (Cysteine)	Cleavable/proteolytic (cathepsin B)	auristatin	cervical cancer

Unprovided data: N/A.

**Table 2 cancers-14-00778-t002:** Preclinical and clinical trials of ADCs in cutaneous lymphoma.

Commercial Name	Target	Percentage of Positivity	Expression in Healthy Tissues	Antibody	Antibody Isotype	Bioconjugation Head (Antibody Amino Acid)	Linker	Drug	Phase
Brentuximab vedotin	CD30	75%	Activated T, B and NK cells	Brentuximab	Chimeric IgG1	Maleimidocaproyl (Cysteine)	Cleavable/proteolytic (cathepsin B)	MMAE	Approved
F0002ADC	CD30	75%	Activated T, B and NK cells	Brentuximab	Chimeric IgG1	Ester NHS (Lysine)	Uncleavable	DM1	Phase I
N/A	Inducible co-stimulator	MF: 61% (*n* = 23), SS: 88% (*n* = 17)	Lymph node, kidney, liver	Mogamulizumab	Murine monoclonal antibody	N/A	N/A	MMAE	Preclinical
N/A	Cell surface heat shock protein 70	N/A	N/A	239-87	N/A	N/A	N/A	MMAE	Preclinical

Unprovided data: N/A.

**Table 3 cancers-14-00778-t003:** Results of brentuximab vedotin clinical trials in cutaneous lymphoma.

Study	Phase	Patient Number	Response Rate	Survival
Duvic et al.	II	48 (28 with MF, 9 with LyP, 2 with pcALCL)	ORR 73% ORR 54% in MF group ORR 100% in other subgroups CR 2/28, PR 13/28 in MF subgroup	PFS 1.1 years (95% CI 0.9−1.4)
Kim et al.	II	32 (MF and SS)	ORR in 21/30 (70%, 90% CI 53−83) CR in 1/30 PR in 20/30SD in 4/30	Median PFS not reached at 12 months Median EFS > 6 months 61% event free at 6 months 28% event free at 12 months
Prince et al.	III Comparison of BV with physician’s choice of either bexarotene or methotrexate	123 (97 with MF, 31 with pcALCL) with 64 in BV group, 64 in PC group	ORR 56.3% (BV group) versus 12.5% (PC group), with *p* < 0.0001 ORR 67% in BV group CR 16% ORR 20% in PC group CR 2%	Median PFS 16.7 months (BV group) versus 3.5 months (PC group), with *p* < 0.0001

Objective response rate: ORR; mycosis fungoides: MF; Sézary syndrome: SS; complete response: CR; partial response: PR; lymphomatoid papulosis: LyP; primary cutaneous anaplastic large cell lymphoma: pcALCL; progression-free survival: PFS; brentuximab vedotin: BV; physician’s choice: PC; event-free survival: EVS.

**Table 4 cancers-14-00778-t004:** Preclinical and clinical trials of ADCs in melanoma.

Commercial Name	Target	Percentage of Positivity	Expression in Healthy Tissues	Antibody	Antibody Isotype	Bioconjugation Head (Antibody Amino Acid)	Linker	Drug	Phase
Glembatumumab vedotin	gpNMB	86% (*n* = 21)	Skin, bones	Glembatumumab	Human IgG2	Maleimidocaproyl (Cysteine)	Cleavable/proteolytic	MMAE	Phase I/II
N/A	PMEL17	64% (*n* = 58)	Melanocytes	17A9	Mouse N/A	N/A	Cleavable/proteolytic	MMAE	Preclinical
EV20/MMAF ADC	HER3	65% (*n* = 130)	Liver, pancreas, epithelial cells	EV20	Humanized IgG1	Maleimidocaproyl (Cysteine)	Uncleavable	MMAF	Preclinical
DEDN6526A	ET_B_R	Majority of tumors (% N/A)	Liver, cortex, medulla	MEDN6000A	Humanized IgG1	Maleimidocaproyl (Cysteine)	Cleavable/proteolytic	MMAE	Phase I
LOP628	c-KIT	66% to 88%	Skin epithelial cells, breast, neurons	LMJ	Humanized IgG1	Ester NHS (Lysine)	Uncleavable	DM1	Phase I (stopped)
*Enapotamab vedotin*	AXL	N/A	Muscle, testis	Enapotamab	Human IgG1	Maleimidocaproyl (Cysteine)	Cleavable/proteolytic	MMAE	Phase I/II

Unprovided data: N/A. gpNMB: glycoprotein-NMB; PMEL17: premelanosome protein 17; HER3: human epidermal growth factor receptor 3; ET_B_R: endothelin B receptor; c-KIT: tyrosine-protein kinase; AXL: Anexelekto.

**Table 5 cancers-14-00778-t005:** Preclinical and clinical trials of ADCs in carcinoma. A. Squamous cell carcinoma. B. Merkel cell carcinoma.

Squamous Cell Carcinoma									
Commercial Name	Target	Percentage of Positivity	Expression in Healthy Tissues	Antibody	Antibody Isotype	Bioconjugation Head (Antibody Amino Acid)	Linker	Drug	Phase
Tisotumab vedotin	TF	75% (*n* = 20)	Brain, heart, intestine, kidney, lung, placenta, uterus	Tisotumab	Human IgG1	Maleimidocaproyl (Cysteine)	Cleavable/proteolytic	MMAE	Phase I/II
Bivatuzumab mertansine	CD44v6	100% (*n* = 5)	Skin keratinocytes, cervix, cornea, tonsil	Bivatuzumab (or BIWA 4)	Humanized IgG1	Disulfide linker SPP (Lysine)	Cleavable/hydrazone	DM1	Phase I (stopped)
Samrotamab vedotin (ABBV-085)	LRRC15	64% (*n* = 115)	Hair follicles, tonsil, stomach, spleen, osteoblasts	Ab1	Humanized IgG1	Maleimidocaproyl (Cysteine)	Cleavable/proteolytic	MMAE	Phase I
**Merkel Cell Carcinoma**									
**Commercial Name**	**Target**	**Percentage of Positivity**	**Expression in Healthy Tissues**	**Antibody**	**Antibody Isotype**	**Bioconjugation Head (Antibody Amino Acid)**	**Linker**	**Drug**	**Phase**
Lorvotuzumab mertansine (IMGN901)	CD56	88% (*n* = 64)	NK cells, neuroendocrine cells, neurons	Lorvotuzumab (huN901)	Humanized IgG1	Disulfide linker SPP (Lysine)	Cleavable/hydrazone	DM1	Phase I
Adcitmer**^®^**	CD56	66% (*n* = 90)	NK cells, neuroendocrine cells, neurons	m906	Human IgG1	Maleimidocaproyl (Cysteine)	Cleavable proteolytic (cathepsin B)	MMAE	Preclincal

TF: tissue factor; LRRC15: leucine-rich repeat containing 15.

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
