# Peer review of "Antibody–Drug Conjugates as an Emerging Therapy in Oncodermatology"

_cancers, 2022, doi:10.3390/cancers14030778_

Round 1

Reviewer 1 Report

This is a well-written review about ADCs as an emerging therapy in oncodermatology. I suggest it for publication after the following points are addressed.

  1. Line 66-69, several recent studies (doi.org/10.3390/molecules26195847; doi.org/10.3390/ph13090245) should be included to support such claim.
  2. A section of perspective is encouraged to be added.

Author Response

We thank the referees, Associate Editor and Referees for their helpful comments which have given us the opportunity to improve our manuscript as suggested. 

Please find the revised version that we hope addresses all concerns of the reviewers. 

Sincerely, 

  1. Esnault
  2. Samimi

Reviewer 1 :

  • Line 66-69, several recent studies (doi.org/10.3390/molecules26195847; doi.org/10.3390/ph13090245) should be included to support such claim.

As requested, both references were added in the paragraph 1.

  • A section of perspective is encouraged to be added.

The paragraph 6 was created named “Strenghts and weakness of ADC: challenges and perspectives”.

We added a simple summary:

Simple summary

Currently the therapeutic arsenal to fight cancers is extensive. Among these, antibody-drug conjugates (ADCs) consist in an antibody linked to a cytotoxic agent, allowing a specific delivery on tumor cells. ADCs are an emerging class of therapeutics with twelve FDA and EMA-approved drugs for hematological and solid cancers. In the recent years, a tremendous progress has been observed in therapeutic approaches for advanced skin cancer patients. ADCs appear as an emerging therapeutic option in oncodermatology. After providing an overview of ADC design and development, the goal of this article is to review the potential ADC indications in the field of oncodermatology.

The bibliography editing was change as requested.

Conflict of interest were added: “AD, CM and MCVM are employees of McSAF, who is the owner of the patent Adcitmer®. AT, TK and MS filed this patent. TK is AD’s husband. The other authors de-clare no conflict of interest.”

Reviewer 2 Report

The manuscript by Esnault et al. provide a comprehensive and well written review regarding antibody drug conjugates with a particular focus in oncodermatological indications. This will offer a synthetic introduction to persons involved in the ADC field and to oncodermatologists.

Here are some minor comments :

  • The number of approved ADCs is a rapidly moving number. Currently there are 12 approved ADCs (7 in onco-hematology and 5 in solid tumors)
  • The authors may want to addess glyco-engineering as a whole, since it may be used both to enhance Fc-mediated mechanisms (to be added line 131) and for ligation purposes (see also van Geel et al., Bioconjugate Chemistry, 2015)
  • In 2.4 it is important to stress the recent approval of topoisomerase 1 inhibitor-containing ADCs, which represents the third family of cytotoxic agents after tubulin-binding agents and DNA-binding agents. Overall however all of these payloads do not differ from conventional cytotoxic chemotherapy in terms of intracellular targets
  • Line 192 : the paradigm that a DAR value of 4 is optimal for pharmacokinetics has been challenged by the recent approval of DAR 8 antibodies, in particular in light of improved hydrophobicity-masking technologies (see Viricel, Chem Sci 2019)
  • Paragraph 2.7 : among the challenges to be dealt with in the ADC field the authors could mention toxicity issues, in particular atypical ocular toxicities, unexpected combinatorial toxicities (bleomycin + BV) (see Honaghy, Mab 2016)
  • Paragraph 4 : titles for subparagraphs are not appropriate since tyrosine kinase receptors are part of membrane proteins (thus 4.1 should not be entitled «membrane protein as targets ») and c-kit is a tyrosine kinase receptor (thus 4.2 is not inclusive of all tyrosine kinase receptors)
  • Figure 3 could be replaced by a more informative table specifiying for each of the four indications : the % of positivity for each target antigen, information concerning expression in healthy tissues, references to preclinical data, references to ongoing clinical trials
  • Add belantamab mafodotin to Table 1A and tisotumab vedotin to Table 1B
  • Table 2 : add other studies concerning Mycosis fungoides/Sezary syndrome/CTCL (see Choudhary et al., ASH 2021 ; Amatore et al. Blood Advances 2021 and others)
  • For clinical trials add NCT identifiers

Author Response

We thank the referees, Associate Editor and Referees for their helpful comments which have given us the opportunity to improve our manuscript as suggested. 

Please find the revised version that we hope addresses all concerns of the reviewers. 

Sincerely, 

C. Esnault

M. Samimi

 Reviewer 2 :

  • The number of approved ADCs is a rapidly moving number. Currently there are 12 approved ADCs (7 in onco-hematology and 5 in solid tumors)

Thank you, the manuscript was corrected to report the twelve approved ADCs.

  • The authors may want to addess glyco-engineering as a whole, since it may be used both to enhance Fc-mediated mechanisms (to be added line 131) and for ligation purposes (see also van Geel et al., Bioconjugate Chemistry, 2015)

As suggested, the concept of glyco-engineering to enhance Fc-mediated mechanisms and for ligation between antibody and drug was added in paragraph 2.6.

  • In 2.4 it is important to stress the recent approval of topoisomerase 1 inhibitor-containing ADCs, which represents the third family of cytotoxic agents after tubulin-binding agents and DNA-binding agents. Overall however all of these payloads do not differ from conventional cytotoxic chemotherapy in terms of intracellular targets

This family of cytotoxic agent was added to the paragraph 2.4.

  • Line 192 : the paradigm that a DAR value of 4 is optimal for pharmacokinetics has been challenged by the recent approval of DAR 8 antibodies, in particular in light of improved hydrophobicity-masking technologies (see Viricel, Chem Sci 2019)

As requested, this point was specified in paragraph 2.6.

  • Paragraph 2.7 : among the challenges to be dealt with in the ADC field the authors could mention toxicity issues, in particular atypical ocular toxicities, unexpected combinatorial toxicities (bleomycin + BV) (see Honaghy, Mab 2016)

These challenges were added to the paragraph 6.

  • Paragraph 4 : titles for subparagraphs are not appropriate since tyrosine kinase receptors are part of membrane proteins (thus 4.1 should not be entitled «membrane protein as targets ») and c-kit is a tyrosine kinase receptor (thus 4.2 is not inclusive of all tyrosine kinase receptors)

Titles of subparagraphs were changed and c-kit was included in the “tyrosine kinase receptor” subparagraph.  

  • Figure 3 could be replaced by a more informative table specifiying for each of the four indications : the % of positivity for each target antigen, information concerning expression in healthy tissues, references to preclinical data, references to ongoing clinical trials

All targets mention in Figure 3 are also present in Tables III, IV and V. The informations about percentage of positivity and expression in healthy tissues are provided in Tables III, IV and V. The reference to preclinical and ongoing clinical trials are available in the article.

  • Add belantamab mafodotin to Table 1A and tisotumab vedotin to Table 1B

Both ADCs were added to the respective Tables.

  • Table 2 : add other studies concerning Mycosis fungoides/Sezary syndrome/CTCL (see Choudhary et al., ASH 2021 ; Amatore et al. Blood Advances 2021 and others)

Studies were added in Table 2 with their description was added in the CTCL paragraph 3.

  • For clinical trials add NCT identifiers

NCT identifiers were added for clinical trials.

We added a simple summary:

Simple summary

Currently the therapeutic arsenal to fight cancers is extensive. Among these, antibody-drug conjugates (ADCs) consist in an antibody linked to a cytotoxic agent, allowing a specific delivery on tumor cells. ADCs are an emerging class of therapeutics with twelve FDA and EMA-approved drugs for hematological and solid cancers. In the recent years, a tremendous progress has been observed in therapeutic approaches for advanced skin cancer patients. ADCs appear as an emerging therapeutic option in oncodermatology. After providing an overview of ADC design and development, the goal of this article is to review the potential ADC indications in the field of oncodermatology.

The bibliography editing was change as requested.

Conflict of interest were added: “AD, CM and MCVM are employees of McSAF, who is the owner of the patent Adcitmer®. AT, TK and MS filed this patent. TK is AD’s husband. The other authors de-clare no conflict of interest.”